# Investigating the role of urban vegetation alongside other environmental variables in shaping *Aedes albopictus* presence and abundance in Montpellier, France

Colombine Bartholomée[1]*, Paul Taconet[1,2], Mathilde Mercat[1], Coralie Grail[1], Emilie Bouhsira[3], Florence Fournet[1], Nicolas Moiroux[1,4]*

**1** MIVEGEC, University of Montpellier, CNRS, IRD, Montpellier, France, **2** TETIS, Inrae, AgroParisTech, CIRAD, CNRS, University of Montpellier, Montpellier, France, **3** InTheres, University of Toulouse, INRAE, ENVT, Toulouse, France, **4** Pôle de Zoologie Médicale, Institut Pasteur de Dakar, Dakar, Sénégal

* colombine.bartholomee@ird.fr (CB), nicolas.moiroux@ird.fr (NM)

## Abstract

Urban greening helps address urbanization challenges, but it may also favor mosquito species, vectors of pathogens causing human diseases. This study examines the relationship between urban vegetation and the presence and abundance of *Aedes albopictus* in Montpellier, the second greenest city of France, while accounting for meteorology, microclimate, air quality, human socio-demography, and landscape. From May to October 2023, adult mosquitoes were collected monthly in urban parks, residential areas, and the highly impervious city center using BG-Pro traps with odor and $CO_2$ attractants. Microclimate data (air temperature and relative humidity) were recorded on-site at each trap location using Hygrobutton data-loggers. Vegetation, land cover, meteorological, air quality, and human demographic data were gathered from open-access databases. *Aedes albopictus* presence and abundance were analysed according to environmental variables taken at different time lags and spatial distances using a two-stage modeling approach: bivariate analyses using generalized linear mixed models were conducted to select variables for inclusion in a multivariate random forest model, aiming to identify the factors that best explain *Ae. albopictus* presence and abundance. While urban vegetation had a limited effect on *Ae. albopictus* presence, the average patch size, and the percentage of area covered by low vegetation were among the most important predictors of abundance. The main predictors for presence were minimum hourly temperature (24h-48h before sampling), minimum atmospheric pressure during sampling, and the weekly cumulated rainfall recorded six weeks before sampling. The most important predictors of abundance were the average patch size of low vegetation, the maximum hourly temperatures during sampling, and the length of roads. To our knowledge, this is the first study examining urban vegetation's influence on *Ae. albopictus* in France, offering

**Data availability statement:** Entomological and environmental data are available from the GBIF database:Bartholomée C, Taconet P, Mercat M, Sutter C, Grail C, Garcia-Marin C, Bouhsira E, Fournet F, Moiroux N (2024). Monthly mosquito sampling in Montpellier (Occitanie), France, 2023. IRD - Institute of Research for Development. Sampling event dataset https://doi.org/10.15468/4qafbu accessed via GBIF.org on 2025-10-02. All scripts are available here: https://archive.softwareheritage.org/swh:1:dir:a1c-83c4be076b5645d1834265219d4f2e6e-5b544;origin=https://github.com/ptaconet/modeling_vector_mtp;vis-it=swh:1:snp:741ed0f1af295a2a2ef477b-310cda3f0faa3f1a0;anchor=swh:1:rev:11de6e23243904d7fa7a21d9d6ade29486d06349.

**Funding:** This work is part of the V2MOC project, led by FF and funded by the Occitanie Region as part of the Défi Clé RIVOC (https://www.umontpellier.fr/universite/projets-em-blematiques/defis-cles-de-la-region-occitanie). CB received a doctoral scholarship from the Défi Clé RIVOC of the Occitanie Region and the University of Montpellier.

**Competing interests:** The authors have declared that no competing interests exist.

insights for urban planning and suggesting further research on vegetation's role in mosquito-borne disease transmission, particularly in the context of increasing dengue incidence in Europe.

## Introduction

According to Hussain and Imitiaz [1], urbanisation is a complex, dynamic, and evolving social process that drives the emergence and growth of cities. It occurs unevenly across time and space, with disparities between continents and countries. Cities are mosaics of heterogeneous patches of diverse ecosystems with multiple interfaces [2]. Urbanisation can improve public health by enhancing living standards for citizens [3]. However, it also results in habitat loss and fragmentation, leading to a decline in biodiversity, as observed in China for birds species diversity [4]. Urbanisation also exacerbates air pollution [5], contributing to human health issues [6]. Moreover, rapid urbanisation often accentuates social disparities, as access to green spaces, healthcare, and healthy living conditions is not equally distributed across urban populations. These inequalities, together with reduced opportunities for contact with nature, increase vulnerability to stress and mental health disorders [7]. Finally, the expansion of asphalt-covered areas that absorb energy during the day and release it as heat at night amplifies the urban heat island (UHI) effect [8], worsening its impact on human health [9].

Given these multiple pressures, strategies that simultaneously address environmental and social challenges are increasingly needed. Nature-Based Solutions (NBS), such as urban greening, are promoted to achieve the 11th Sustainable Development Goal: "to make cities and human settlements inclusive, safe, resilient, and sustainable" [10]. NBS also support the 12th target of the COP 15 Biodiversity Plan for Life on Earth, aiming to "enhance green spaces and urban planning for human well-being and biodiversity" [11], and align with the sustainability goals of the European Green Deal [12]. Urban greening involves deliberately integrating vegetation into urban areas [13], encompassing initiatives such as soil de-imperviousness, developing green and blue infrastructures, conserving urban green spaces (UGS), and promoting urban agriculture. UGS provide cultural and regular ecosystem services [13]. They help mitigate the UHI effect [14,15], and contribute to improving mental health [16]. Effective management of UGS can help sequester carbon dioxide [17] through tree plantations and enhance animal biodiversity by increasing the presence of indigenous vegetation [18].

Urban greening may also have negative side effects. It can increase pollen allergies [19] and attract agricultural pests like tree moths [20]. UGS increase human-wildlife interactions, raising the risk of zoonotic [21] and arthropod-borne diseases [22], as shown for the Lyme disease pathogen (*Borrelia burgdorferi*) transmitted by ticks in urban parks across Europe [23]. In Europe, while numerous studies have explored the relationship between UGS and tick populations, the relationship between urban greening and mosquitoes is less documented [24], with only one study on *Culex pipiens* in the Netherlands [25] and two studies on the tiger mosquito *Aedes albopictus* in Italy [26,27]. As shown in the scoping review by Mercat *et al*

(2025) [24], the effects of urban green infrastructures (UGIs) on mosquito populations are highly dependent on the vector system studied and the environmental context. Urban greening is a dynamic process, and its environmental changes may take time to manifest [28]. While it can create favorable conditions for vectors and increase human-vector interactions, it may also promote the establishment of predators that could regulate vector populations.

*Aedes albopictus* is an invasive mosquito in Europe. It is native to Southeast Asia and a vector of several infectious agents, including the dengue virus (DENV), chikungunya virus (CHIKV), and Zika virus. The species adapted to anthropogenic environments [29], including urban areas, using natural and artificial breeding sites. It can survive extreme conditions through egg diapause [30]. Its abundance is influenced by environmental factors such as weather [31], microclimate [32], and landscape features [33]. Urban vegetation may favour *Ae. albopictus* by creating cooler and more humid microclimates during warm season [34], facilitating access to sugar meals, providing natural resting sites [35], and attracting blood sources (humans and animals). In France, since its first detection in 2004 [36], *Ae. albopictus* spreads to 78 of 96 metropolitan France's departments by 2024, including all departments in the Occitanie region, where it has been linked to several autochthonous arboviral disease clusters since 2014.

The Occitanie region is actively pursuing urban greening projects in its two largest cities, Toulouse and Montpellier, including plans to plant 150 000 trees. As Occitanie is a hotspot for both arboviral diseases and urban greening, it is crucial to assess the impact of these greening initiatives on *Ae. albopictus* population dynamics. With the goal to guide future research to improve the design and planning of urban green spaces, this study aims to evaluate the influence of urban vegetation in Montpellier in 2023 on the presence and abundance of *Ae. albopictus* females, using an explanatory approach taking into account land cover, microclimate, meteorology, air quality, and socio-demographics.

## Materials and methods

### Mosquito sampling and study areas

**Study areas.** Montpellier is a city of 300 000 inhabitants located in the Occitanie region in southern France (Fig 1). The city has a Mediterranean climate with an average annual temperature of 15°C and an average annual precipitation of 739 mm [37]. The city features more than 10% of vegetated areas within densely urbanized areas. It ranks among the top three greenest cities in metropolitan France [38]. The study was conducted in nine areas belonging to three different environments: two urban parks (PRK), three residential neighborhoods (RES), and four highly impervious (i.e., areas of the city center dominated by sealed surfaces such as concrete and asphalt) areas (IMP) (Fig 1). Urban parks (Botanical Garden (PRK-BOT), and Park of Aiguelongue (PRK-AGL)) and residential neighborhoods (Lemasson (RES-LEM), Aiguerelles (RES-AGR), and Soulas (RES-SOUL)) were vegetated areas. The highly impervious areas included two areas with some vegetation (Saint Charles University (IMP-SCU) and Acapulco Hotel (IMP-ACA)) and two with little to no vegetation (Buisson Bertrand Institute (IMP-BBI) and Diderot School (IMP-DID)) (Fig 1). Detailed descriptions of study areas are provided in S1 Table, and images of study areas in S1 Fig. The minimum distance between study areas was 115 m. Although *Aedes* mosquitoes can disperse from 75 m [39] to 290 m [40] in urban areas over 8 days, we treated the study areas as independent units due to impervious roads between them, that are known to limit mosquito dispersal [41,42].

**Mosquito sampling.** Adult mosquitoes were captured using BG-Pro traps (BioGents, Regensburg, Germany) modified according to L'Ambert *et al* [43], each equipped with a BG lure—synthetic human odor mimic—and a daily replenished carbon dioxide source made from yeast, sugar, and water (S1 Fig). Two BG-Pro traps were installed in each residential area (RES-LEM, RES-AGR, RES-SOUL), and in the park of Aiguelongue (PRK-AGL). Due to their smaller size (S1 Table), only one BG-Pro trap was deployed in each of the highly impervious areas (IMP-ACA, IMP-BBI, IMP-DID and IMP-SCU). Because of its larger size (S1 Table), three BG-Pro traps were deployed in the Botanical Garden (PRK-BOT).

**A. Location of Montpellier**

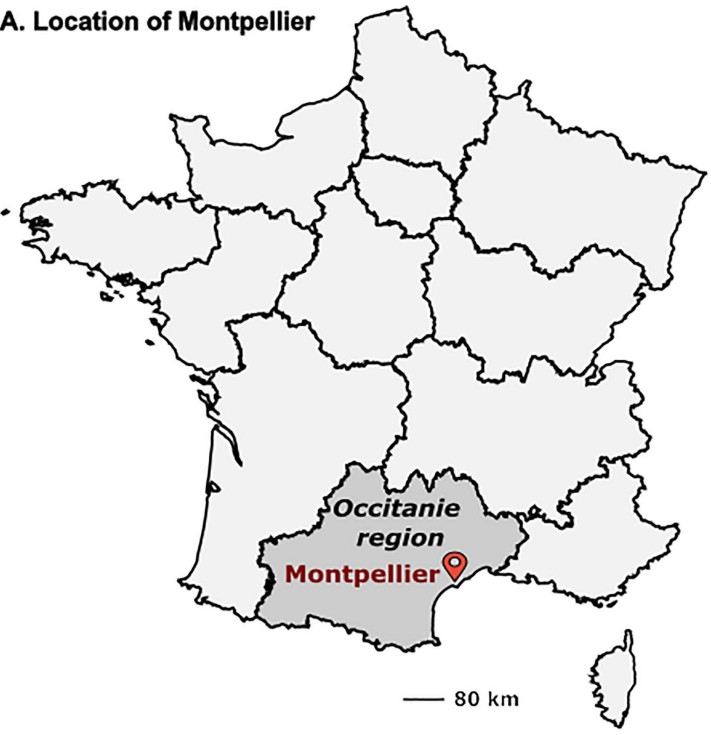

**C. Residential area**

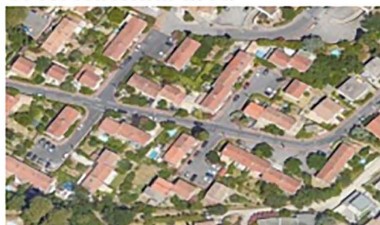

**D. Botanical Garden**

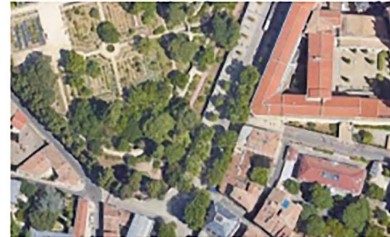

**E. One impervious area**

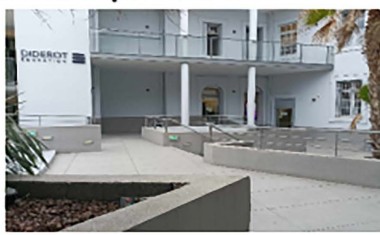

**B. Studied areas**

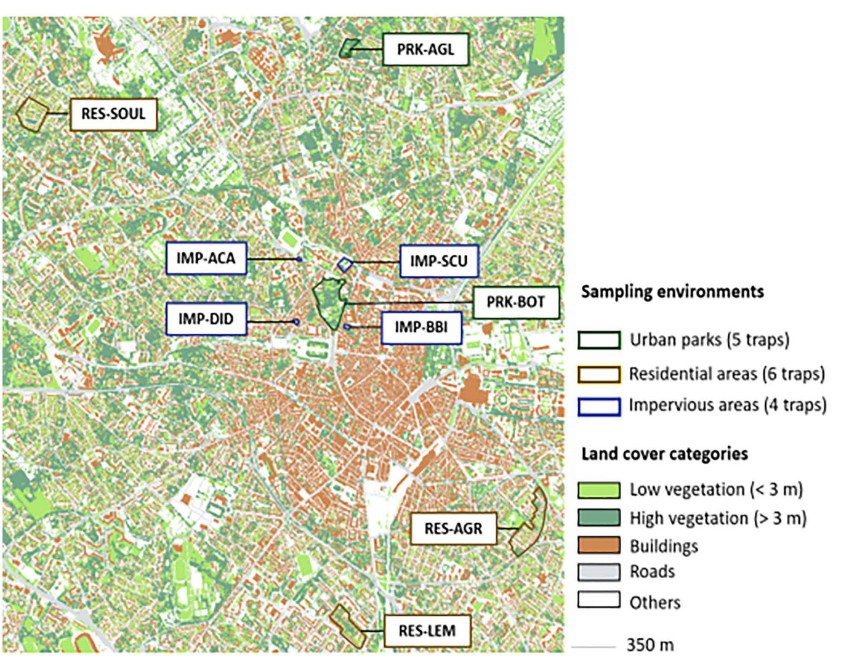

**Sampling environments**

- Urban parks (5 traps)
- Residential areas (6 traps)
- Impervious areas (4 traps)

**Land cover categories**

- Low vegetation (< 3 m)
- High vegetation (> 3 m)
- Buildings
- Roads
- Others

**Fig 1. Study areas.** Fig 1A and 1B show the location of Montpellier in France (map from Map data © OpenStreetMap contributors, licensed under ODbL) and the land cover map of Montpellier with the study areas, respectively. Sampling environments are shown in the legend of Fig 1B, delineating urban parks, residential areas, and highly impervious areas (boundaries from Montpellier Open Data). Land cover categories are shown in the legend of Fig 1B. Fig 1C, 1D, and 1E show a garden of a residential area, the Botanical Garden (PRK-BOT), and one highly impervious area respectively. Photographs and image created by the author (Colombine Bartholomée).

Traps were placed in shaded, wind-sheltered sites at a height of one meter. Therefore, 15 sites were sampled: five in the two urban parks, six in the three residential areas, and four in highly impervious areas. Every trap was separated by a minimum distance of 50 m to prevent interferences [44]. However, due to accessibility constraints, the two traps in RES-SOUL were placed 22 m apart.

One 48-hour trapping session was conducted per site per month from May to October 2023. During each session, mosquitoes were retrieved each morning, resulting in two collections per session. Mosquitoes were transported at 4°C to the laboratory (MIVEGEC laboratory, IRD regional delegation, Montpellier), and stored at −20°C. Mosquito individuals were then counted, sorted by sex, and morphologically identified using the multicriteria key MosKeyTool [45]. These data are available on GBIF [46].

## Environmental data collection and preparation

To assess the impact of urban vegetation on the presence and abundance of *Ae. albopictus*, it was essential to consider all factors that may influence its population dynamics. For this task, we adapted the conceptual model of human-*Anopheles* contact by Moiroux (2012) [47] to the case of *Ae. albopictus*. Based on a literature review, we documented and hypothesised known or potential predictors and their pathways to mosquito abundance [48]. This model guided the selection of environmental data to be collected as described below.

**Fine-scale vegetation and land cover data.** A harmonised land-cover map of Montpellier was generated at 0.5 m resolution by combining three complementary datasets (see Supplementary S2 Table): (i) the fine-scale vegetation dataset of Montpellier (Montpellier Méditerranée Métropole, 2019), which distinguishes vegetation patches using deep learning and photo-interpretation of Pléiades satellite imagery and a LiDAR-derived elevation model; (ii) the French national building database (Base de données nationale des bâtiments, BDNB); and (iii) the Copernicus Urban Atlas. Following rasterisation and integration of these sources, five land-cover categories were retained (Fig 1B): buildings, roads, low vegetation (< 3 m height), high vegetation (> 3 m height), and "others" (including railways, pathways, industrial areas, and sports facilities).

For each trap location, spatial metrics were computed within circular buffer zones of 20 m, 50 m, 100 m, and 250 m radii. For each land cover category, we calculated the percentage cover (proportion of buffer area occupied by the category), total edge length (sum of the perimeters of all patches of the category), and average patch size (mean area of individual patches). For vegetation categories (low vegetation < 3 m height, high vegetation > 3 m height), we also extracted the number of patches. The landscape diversity (Shannon index) was calculated for all buffer zones. This index quantifies diversity by considering both the number of land cover categories and their relative proportions, with higher values indicating more heterogeneous landscapes (see Supplementary S2 Table for the formula). Buffer sizes were chosen according to previous works investigating the effect of urban vegetation [27,49] on the dispersal of *Ae. albopictus* [40]. Supplementary S3 Table provides the land cover variable values for each trap and buffer radii.

**Microclimatic data.** We continuously collected hourly temperature and relative humidity near each trap using Hygro Button data loggers from May to October 2023. The minimum (lowest recorded value), maximum (highest recorded value), and mean temperatures, as well as relative humidity, were recorded for 24-hour periods starting from 48 hours prior to the beginning of each sampling session and continuing until the end of the session (S2 Table).

**Meteorological and air quality data.** Hourly temperature and rainfall data were obtained from the Observatoire Départemental de l'Eau et de l'Environnement (ODEE) station located at Château d'Ô in the north-west of Montpellier (S2 Table). Relative humidity and wind speed data were obtained from the Frejorgues airport station (belonging to the Meteo France national meteorology agency network) at a 3-hour resolution (S2 Table). Growing Degree Days (GDD), a known predictor of mosquito development, were calculated from daily mean temperatures above 11°C, as this was shown to be the annual mean temperature limit for the establishment of *Ae. albopictus* populations [50]. Temperatures above this level are conducive to mosquito development. For each sampling event, weather data were aggregated at a weekly resolution

for the sampling week (W0) and for all possible lag periods between W–6 and W0. These lag periods ranged in duration from one to seven consecutive weeks (e.g., W–2 only, W–4 to W–2, W–6 to W0). Rainfall and GDD were summed, while temperature, humidity, and wind speed were averaged over each lagged window. Precipitation data from the Château d'Ô station were also aggregated over 24-hour periods prior to and during each sampling session, following the same temporal aggregation approach used for microclimate data from the data loggers, because precipitation may influence mosquito activity and therefore sampling.

As atmospheric pressure can influence insects' flight and activity [51–53], we evaluate the effect of daily atmospheric pressure on *Ae. albopictus*. For this purpose, the minimum (lowest recorded value), maximum (highest recorded value), and mean atmospheric pressure, measured every three hours at the Frejorgues airport station, were extracted per 24-hour periods before and during sampling sessions. The daily mean atmospheric pressure difference between two consecutive days was calculated per 24-hour periods before and during sampling sessions. S3 Fig depicts the temporal evolution of these meteorological variables (temperature, rainfall, wind speed, relative humidity, and atmospheric pressure).

As air pollution can influence insect behaviour and abundance [54–56], we evaluate the effect of air quality data on *Ae. albopictus*. For each trap location, we collected daily concentrations of nitrogen monoxide (NO), nitrogen dioxide (NO2), particulate matter (of diameter less than 10 (PM10) and 2.5 μm (PM2.5)), and ozone (O3) from the closest station of the Atmo Occitanie air quality observatory (S2 Table). As for weather data, air quality data were aggregated at a weekly resolution for the sampling week (W0) and for all possible lag periods between W–6 and W0, by averaging the daily concentrations of each pollutant over these windows.

**Social and demographic data.** Socioeconomic factors such as low living standards [57], aging or degrading infrastructure [58], and population densities [59] can influence mosquito presence. Number of buildings of different ages and number of low-income households (2019) were obtained from the Filosofi database of the French Institut National de la Statistique et des Études Économiques (INSEE), provided as a 200 m resolution grid. For each buffer area, values of intersecting grid cells were summed. Human population data were obtained from the Montpellier fine-scale population dataset of 2020 (Supplementary S2 Table), in which IRIS-level population counts (~2000 inhabitants per unit) were redistributed across land plots in proportion to living area and assigned to the centroid of significant buildings. Population counts were then aggregated within the same four buffer zones (20, 50, 100, and 250 m radii).

**Assessment of vector control interventions.** No municipal adulticide or larvicide treatments were implemented in the study areas during the sampling period. No individual larval habitat removal by residents was observed. However, potential actions by residents near some sampling sites could not be systematically assessed, and their effects on mosquito abundance cannot be excluded.

## Statistical analyses of the presence and abundance of female *Ae. albopictus*

To study the relationships between the presence and the abundance of female *Ae. albopictus* and environmental conditions, we analysed the data using a hurdle modeling approach, i.e., considering the data responding to two distinguished processes: one causing zero vs. non-zero (« presence » models) and one causing positive counts of females (« abundance » models). This approach is justified because determinants of mosquito presence are expected to be different than abundance ones [60].

For both presence and abundance analysis, we adopt a two-stage analysis method as described by Taconet *et al* [61]. The first step involves bivariate analyses using Generalized Linear Mixed Models (GLMMs), followed by the second step, a multivariate analysis employing an interpretable machine learning method based on random forest models [62]. The bivariate analyses aimed to improve our understanding of (i) the time-lagged effects of meteorological and air quality variables, (ii) the impact of atmospheric pressure and microclimate on the tiger mosquito, (iii) land cover-scale effects, (iv) effects of socio-economic and demographic variables, and (iv) to identify and select key variables for multivariate analysis. The multivariate analysis aimed to identify the most impactful determinants of the presence and abundance of

*Ae. albopictus* and the shape of their association while considering potentially complex (e.g., non-linear) associations and interactions between variables, with the overall aim of assessing more precisely the impact of urban vegetation. The modeling workflow is summarised in S2 Fig and in S1 Text.

**Bivariate analysis using GLMMs.** Every explanatory variable was tested using a GLMM with crossed random intercepts for trap location, with sites nested within the sampling area, and for the sampling session. We fitted binomial and zero-truncated negative binomial models to presence/absence and positive counts data, respectively. Marginal R2, representing the proportion of variance explained by the fixed effects excluding the contribution of random intercepts [63], was calculated and used for interpretation. Time-lagged effects of meteorological and air quality variables were interpreted using cross-correlation maps (CCMs) [64] of marginal R2 adjusted for the direction (+/-) of the relationship.

**Multivariate analysis using random forests.** The results of the bivariate analyses were used to select the variables to be included in the multivariate analysis. First, we excluded variables poorly correlated with the response variables (p > 0.2). For each meteorological, microclimatic, air quality, land cover, and demographic variable, we retained the time lag or radii with the higher marginal R2. Multicollinearity was addressed using Pearson's correlation coefficient with a threshold of 0.7. Then, based on empirical knowledge, an initial selection of correlated variables was made. Other collinear covariates were removed until the Variance Inflation Factor (VIF) of the remaining variables was below three [65] (S2 Fig and S1 Text). Binary classification random forests (RFs) and regression classification RFs were generated for the presence and abundance models, respectively. To improve model efficiency and reduce the number of explanatory variables, recursive feature elimination was performed for each model [66]. To account for spatial and temporal autocorrelation, sampling variables (trap location, sampling day, and sampling session) were included. For each set of hyperparameters and to assess the predictive power of each model, both a spatial and temporal cross-validation framework [67] was used, taking into account the potential spatial and temporal autocorrelation in the data. Spatial cross-validation (leave-area-out) involved training the model recursively on data from eight out of the nine sampled areas, testing it with data from the remaining area, and finally averaging the performance metric for the nine areas [61]. The temporal cross-validation (leave-sampling-session-out) involved training the model recursively on data from six out of the seven sampling sessions and testing it with data from the remaining session. The performance metrics selected were the Area Under the Curve (AUC) for the presence models [68] and the Mean Absolute Error (MAE) for the abundance models. The model results were interpreted using Variable Importance Plots (VIPs) [62] and Partial Dependence Plots (PDPs) [69]. Residual spatial autocorrelation was assessed for each model.

**Softwares used.** The R version 4.2.3 programming language [70] was used. The land cover map creation and landscape metrics calculation involved the use of the 'sf' [71], 'raster' [72], 'fasterize' [73], and 'landscapemetrics' [74] packages. GLMMs were fitted using the 'glmmTMB' package [75]. Marginal R2 was calculated using the 'performance' package [76]. The 'correlation' package was used to evaluate multicollinearity [77]. Multivariate analyses were performed using the 'caret' [78] and 'ranger' [79] packages. The 'CAST' package [80] was used for the cross-validation and the interpretation plots were performed using the 'iml' [81], 'pdp' [82], 'MLmetrics' [83], 'precrec' [84] and 'patchwork' [85] packages. The package 'spatialRF' [86] was used to assess spatial autocorrelation. All scripts are available on Software Heritage [87].

## Results

### Entomological data

A total of 2344 mosquitoes were collected, predominantly *Ae. albopictus* (85.66%, N = 2008), with smaller proportions of *Culex pipiens* (13.91%, N = 326), *Culiseta longiareolata* (0.39%, N = 9), and *Culiseta annulata* (0.04%, N = 1). Among 2008 collected *Ae. albopictus*, there were 1106 females (55.08%) and 902 males (44.92%). The mean density of female *Ae. albopictus* was 3.83 mosquitoes/trap/24h (SE = 1.97) in urban parks, 3.31 mosquitoes/trap/24h (SE = 1.65) in residential

areas, and 1.99 mosquitoes/trap/24h (SE = 1.08) in highly impervious areas (Table 1). Mean mosquito densities (mosquitoes/trap/24 h), stratified by sampling environment (or sampling areas), are presented in Table 1 (or S4 Table). The temporal distribution of *Ae. albopictus* (S4 Fig) exhibited a bimodal pattern, with an initial emergence in May, a primary peak in late June, and a secondary, smaller peak in late August.

### Bivariate analysis

**Vegetation and other land cover variables.** Fig 2 shows the vegetation and other land cover variables correlated with the probability of presence or abundance of female *Ae. albopictus*. Two variables related to low vegetation were significantly and positively associated with the probability of presence within a 20 m radius buffer around the traps: the total length (in m) of low vegetation edges (OR =1.007, $R^2$ = 0.042, p = 0.042, CI 95% [1.001; 1.014]) and the average patch size (in m2) of low vegetation (OR =1.01, $R^2$ = 0.07, p = 0.006, CI 95% [1.003; 1.017]). The percentage of area occupied by buildings within a 50 m radius buffer around the traps was negatively correlated with the probability of presence (OR =0.959, $R^2$ = 0.047, p = 0.048, CI 95% [0.920; 0.999]). Within a 20 m radius buffer around the trap, the total length edge of roads and the average size of a patch of roads were negatively correlated with the probability of presence (Fig 2).

Regarding abundance, positive associations were observed with all vegetation-related variables except for the average patch size of low vegetation within a 50 m radius buffer that showed a negative and significant association (RR = 0.996, $R^2$ = 0.098, p = 0.0001, CI 95% [0.994; 0.998]). The most remarkable positive association was for the total edge length of low vegetation within a 100 m buffer (RR = 1.0008, $R^2$ = 0.190, p = 0.007, CI 95% [1.0002; 1.0014]). Variables representing the building land cover category consistently showed negative relationships with female *Ae. albopictus* abundance, especially for the percentage area variable (across all buffer radii) and for all variables within the 100m buffer radius. RR for the percentage of building areas in the 100m buffer was 0.945 ($R^2$ = 0.240, p = 0.0007, CI 95% [0.914; 0.976]) (Fig 2).

**Microclimatic, meteorological, and air quality variables.** Fig 3 shows the relationship between presence or abundance and microclimatic or meteorological variables recorded from 48h before the start of sampling and during sampling. Local relative humidity (RH) variable showed positive associations with presence when recorded before sampling (with the strongest association for maximum hourly relative humidity 24-48h before sampling; OR = 1.194, $R^2$ = 0.201, p = 0.002, CI 95% [1.070; 1.332]) and negative association when recorded during sampling (with the strongest association for minimum RH; OR =0.924, $R^2$ = 0.134, p = 0.001, CI 95% [0.880;0.969]). Significant associations of RH data with abundance were negative whatever the period considered.

**Table 1. Mosquito density/trap/24h by species and sex in the three different sampling environments.**

| Environment | Mosquito female /trap/24h (± SE) | Mosquito male /trap/24h (± SE) | *Ae. albopictus* female /trap/24h (± SE) | *Ae. albopictus* male /trap/24h (± SE) | *Cx. pipiens* female/ trap/24h (± SE) | *Cx. pipiens* male/ trap/24h (± SE) | *Cs. longiareolata* female /trap/24h (± SE) | *Cs. longiareolata* male /trap/24h (± SE) | *Cs. annulata* male /trap/24h (± SE) |
|---|---|---|---|---|---|---|---|---|---|
| Urban parks | 6.46 (± 2.80) | 3.59 (± 2.89) | 3.83 (± 1.97) | 1.83 (± 1.67) | 1.48 (± 0.79) | 0.49 (± 0.38) | 0.02 (± 0.02) | 0.02 (± 0.02) | 0.01 (± 0.01) |
| Residential areas | 3.97 (± 1.64) | 1.58 (± 1.14) | 3.31 (± 1.65) | 0.93 (± 0.83) | 0.30 (± 0.16) | 0.07 (± 0.05) | 0.01 (± 0.01) | 0.01 (± 0.01) | 0 |
| Highly impervious areas | 2.91 (± 1.37) | 1.19 (± 0.87) | 1.99 (± 1.08) | 0.72 (± 0.69) | 0.26 (± 0.18) | 0.16 (± 0.12) | 0.01 (± 0.01) | 0.01 (± 0.01) | 0 |

SE: Standard error.

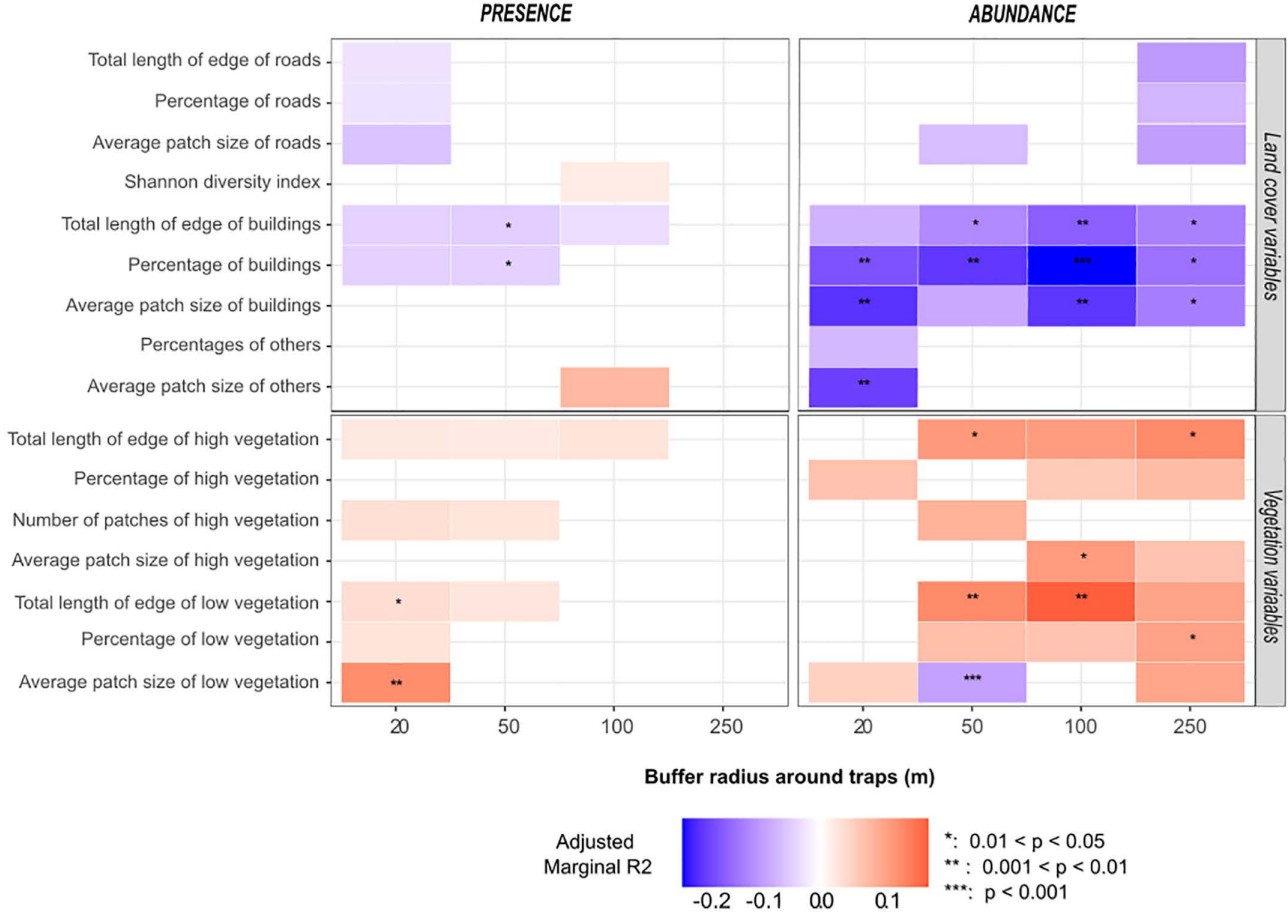

**Fig 2. Bivariate relationships between *Ae. albopictus* and land cover variables for varied buffer sizes.** Relationships are analysed based on *Ae. albopictus* presence or abundance and land cover variables for different buffer sizes around traps. The adjusted marginal R2 reflects the variance explained by the explanatory variable, adjusted for correlation direction. Boxes are colored if the p-value was < 0.2 (No asterisk: p-value ∈ [0.05; 0.2], *: p-value ∈ [0.01, 0.05], **: p-value ∈ [0.001; 0.01], ***: p-value ∈ [0; 0.001]). Box color depends on the direction of the relationship (blue: negative, red: positive). Color intensity varies according to the marginal R2 value.

Local temperatures during and before sampling consitently showed positive correlations with presence, particularly for mean and maximum hourly temperatures recorded during sampling (tmean: OR =1.592, R2 = 0.408, p = 0.0002, CI 95% [1.251; 2.026]; tmax: OR =1.497, R2 = 0.402, p = 0.0001, CI 95% [1.226; 1.828]). Abundance was also positively associated with tmean (RR = 1.143, R2 = 0.197, p = 0.0002, CI 95% [1.064; 1.228]) and tmax (RR = 1.057, R2 = 0.061, p = 0.07, CI 95% [0.996; 1.122]) during sampling, but with lower R2.

Daily total precipitation was negatively associated with presence during sampling (OR =0.863, R2 = 0.137, p = 0.003, CI 95% [0.783; 0.951]). *Aedes albopictus* presence was positively associated with all variables describing atmospheric pressure recorded during the sampling hours, but negatively with lagged data. The strongest effect size was for the difference in mean daily atmospheric pressure (Patdiff) between the sampling day and the previous day (OR = 1.008, R2 = 0.194, p = 0.008, CI 95% [1.002; 1.014]). Average (patmean) and maximum (patmax) atmospheric pressure during the 24-48h period preceding sampling showed negative association with abundance (RR = 0.997, R2 = 0.243, p = 0.018, CI 95% [0.994; 0.999] and RR = 0.998, R2 = 0.177, p = 0.0001, CI 95% [0.996; 0.999], respectively).

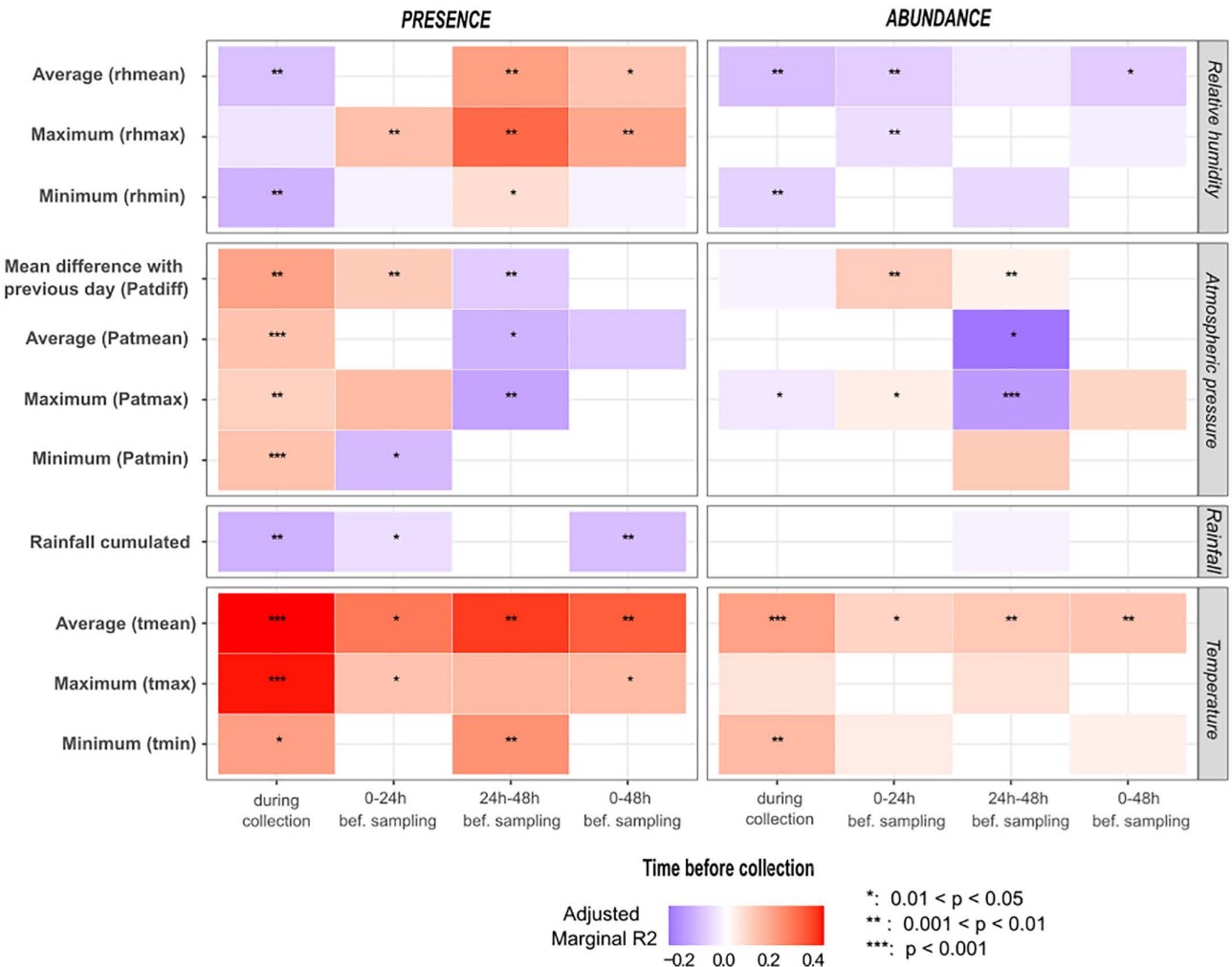

**Fig 3. Bivariate relationships between *Ae. albopictus* and microclimatic and meteorological variables 48h before and during sampling.**
Relationships are analysed based on *Ae. albopictus* presence or abundance and variables recorded 48h before the start and during sampling sessions. Microclimatic variables include minimum hourly relative humidity (rhmin), maximum hourly relative humidity (rhmax), mean hourly relative humidity (rhmean), minimum hourly temperature (tmin), maximum hourly temperature (tmax) and mean hourly temperature (tmean). Meteorological variables include daily cumulated rainfall, minimum atmospheric pressure (Patmin), maximum atmospheric pressure (Patmax), mean atmospheric pressure (Patmean), and the difference of mean atmospheric pressure between two consecutive days (Patdiff). The adjusted marginal R2 reflects the variance explained by the explanatory variable, adjusted for correlation direction. Boxes are colored if the p-value is < 0.2 (No asterisk: p-value $\in$ [0.05; 0.2], *: p-value $\in$ [0.01, 0.05], **: p-value $\in$ [0.001; 0.01]; ***: p-value $\in$ [0; 0.001]). Box color depends on the direction of the relationship (blue: negative, red: positive). Color intensity varies according to the marginal R2 value. bef=before.

Local temperatures during and before sampling consitently showed positive correlations with female presence, particularly for mean and maximum hourly temperatures recorded during sampling (tmean: OR =1.662, R2 = 0.439, p = 0.0001, CI 95% [1.298; 2.129]; tmax: OR =1.553, R2 = 0.424, p = 0.0001, CI 95% [1.262; 1.991]). Abundance was also positively associated with tmean (RR = 1.134, R2 = 0.197, p = 0.0001, CI 95% [1.055; 1.218]) and tmax (RR = 1.052, R2 = 0.050, p = 0.098, CI 95% [0.991; 1.117]) during sampling, but with lower R2.

Daily total precipitation was negatively associated with presence during sampling (OR =0.859, R2 = 0.133, p = 0.003, CI 95% [0.777; 0.949]). *Aedes albopictus* presence was positively associated with all variables describing atmospheric

pressure recorded during the sampling hours, but negatively with lagged data. The strongest effect size was for the difference in mean daily atmospheric pressure (Patdiff) between the sampling day and the previous day (OR = 1.008, R2 = 0.217, p = 0.003, CI 95% [1.003; 1.015]). Average (patmean) and maximum (patmax) atmospheric pressure during the 24-48h period preceding sampling showed negative association with abundance (RR = 0.997, R2 = 0.217, p = 0.049, CI 95% [0.994; 0.999] and RR = 0.998, R2 = 0.183, p = 0.0003, CI 95% [0.996; 0.999], respectively).

Fig 4 shows the effect size and significance (p < 0.2) of weeks-lagged meteorological variables on the presence and abundance of female *Ae. albopictus*. The relationship between weekly cumulated rainfall (CUMRF) and both the probability of presence and abundance was positive, with a statistically significant association for abundance and the highest effect size observed the 6th week before sampling (for presence: OR = 1.430, R2 = 0.334, p = 0.168, CI 95% [0.860; 2.375]; for abundance: RR = 1.075, R2 = 0.127, p = 0.04, CI 95% [1.003; 1.152]). The probability of presence was positively and significantly associated with minimum (TMIN), average (TMEAN), and maximum (TMAX) temperatures. The highest effect size was observed for TMIN one week before sampling (OR =1.620, R2 = 0.372, p = 0.001, CI 95% [1.209; 2.172]. For abundance, relationships with TMIN, TMEAN, and TMAX were also positive and significant for time interval 0–2 weeks before sampling, with the highest effect size observed for TMAX (RR = 1.161, R2 = 0.192, p = 0.0006, CI 95% [1.067; 1.267]). Both responses were positively and significantly correlated with the weekly growing degree days (GDD). The highest effect on presence and abundance was observed for GDD during the week preceding sampling (OR = 1.085,

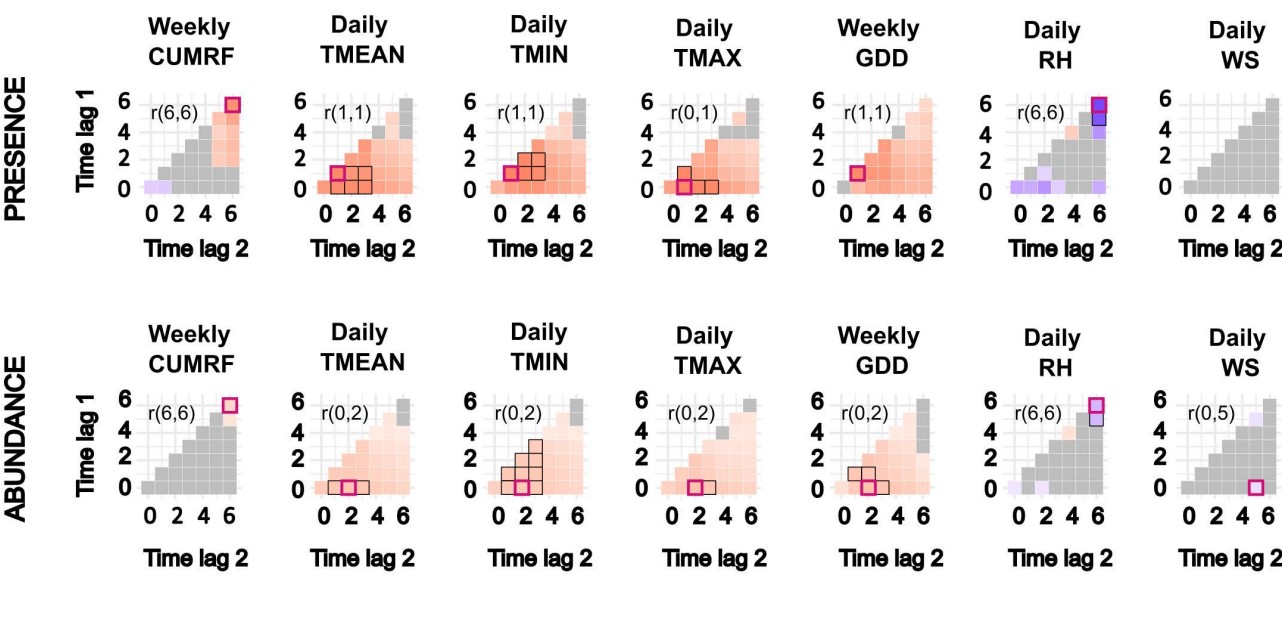

**Fig 4. Cross correlation maps showing the lagged effect of meteorological variables on *Ae. albopictus* presence or abundance.** Lagged meteorological variables include weekly cumulated daily rainfall (CUMRF), mean daily temperature (TMEAN), daily minimal temperature (TMIN), daily maximal temperature (TMAX), the weekly Growing Degree Days (GDD), mean daily relative humidity (RH), and mean daily wind speed (WS). Time lags are expressed in weeks prior to sampling. The adjusted marginal R2 reflects the variance explained by the explanatory variable, adjusted for correlation direction. Red-bordered squares highlight the time lag with the highest marginal R2, with the interval indicated in the top left corner. Black-bordered squares denote marginal R2 values close to the highest (within 10%). Gray squares represent correlations with p-values > 0.2.

R2 = 0.357, p = 0.002, CI 95% [1.031; 1.141]) and for time interval 0–2 weeks before sampling (RR = 1.011, R2 = 0.189, p = 0.0009, CI 95% [1.004; 1.017]), respectively. Both responses were negatively and significantly correlated with relative humidity (RH), with the highest effect observed for RH recorded 6 weeks before sampling (OR = 0.792, R2 = 0.472, p = 0.006, CI 95% [0.670; 0.937] for presence; RR = 0.948, R2 = 0.183, p = 0.007, CI 95% [0.912; 0.986] for abundance). The relationship between wind speed (WS) and abundance was negative, with the highest effect observed for time interval 0–5 weeks before collection (RR = 0.343, R2 = 0.086, p = 0.157, CI 95% [0.078; 1.510]).

Daily nitrogen dioxide (NO2) concentrations were positively associated with the probability of presence of female *Ae. albopictus*, with the highest effect size observed for the sampling week (OR = 1.109, R2 = 0.126, p = 0.163, CI 95% [0.959; 1.282]; S5 Fig). Particulate matter concentrations (PM2.5 and PM10) were also positively correlated with the probability of presence and with the abundance. However, PM2.5 and PM10 concentrations were strongly correlated with all time-lagged temperature metrics (Pearson coefficient > 0.8). Tropospheric ozone (O3) concentration (three weeks before sampling) was negatively correlated with the probability of presence and positively (one week before sampling) with abundance. Daily O3 concentrations were also highly correlated with weekly cumulated rainfall, wind speed, and daily relative humidity (Pearson coefficient > 0.8).

**Socio-demographic variables.** Human population density in a 250 m radius buffer around traps was positively correlated with the *Ae. albopictus* probability of presence (OR = 1.0013, R2 = 0.056, p = 0.029, CI 95% [1.0001; 1.0026]; S6 Fig).

## Multivariate analysis

After bivariate analyses, multicollinearity assessments, and recursive feature elimination, ten and nine variables, in addition to sampling design variables, were kept for the multivariate random forest models of presence and abundance, respectively (S5 Table shows reason for exclusion). Figs 5 and 6 display the multivariate RF model interpretation plots for presence and abundance, respectively. Figs 5A and 6A illustrate the contribution of each variable in explaining the response variable (VIPs). Figs 5B and 6B depict the relationship between the response variable and the explanatory variables (PDPs). Models of presence and abundance did not show any residual spatial autocorrelation.

**Presence model.** The most contributing variables to presence probability were a combination of real-time and time-lagged microclimatic and meteorological variables (Fig 5A), with only a minimal contribution of vegetation variables. The first contributing variable was the local minimum hourly temperature recorded 24–48 hours before sampling. The probability of presence showed a slight increase between 11.9°C and 16°C, rose sharply from 16°C to 25°C, and plateaued beyond 25°C (Fig 5B). Minimum atmospheric pressure during sampling was the second most influential variable, with a unimodal relationship: presence probability increased up to 1012.91 HPa before declining. Cumulated rainfall 6 weeks before sampling emerged as the third key variable, displaying a linear positive association with presence probability, from 0 mm to 15 mm. The fourth major factor was the mean daily atmospheric pressure difference between the sampling and previous day. Presence probabilities decreased for descreasing pressures and were stable for stable or increasing pressures. Secondary contributors included mean local hourly relative humidity 24–48 hours before sampling, daily NO$_2$ concentrations, and vegetation metrics within a 20 m buffer around traps (average patch size of low vegetation, number of high vegetation patches, and total length of low vegetation edges). These variables slightly increased presence probability, whereas the total length of edges of roads in a 20 m radius buffer weakly reduced it. The model demonstrated strong predictive accuracy (AUC = 0.89; S7A Fig). Cross-validation with the leave-area-out method produced consistent predictions, except in the Diderot School area (IMP-DID). In contrast, leave-sampling-session-out cross-validation exhibited greater variability between sampling areas (S7B Fig).

**Abundance model.** The most contributing variables to *Ae. albopictus* abundance comprised a combination of microclimatic, weeks-lagged meteorological, vegetation and other land cover variables. The first contributing variable was the average size of low-vegetation patches within a 50 m radius buffer around traps (Fig 6A). Abundance increased

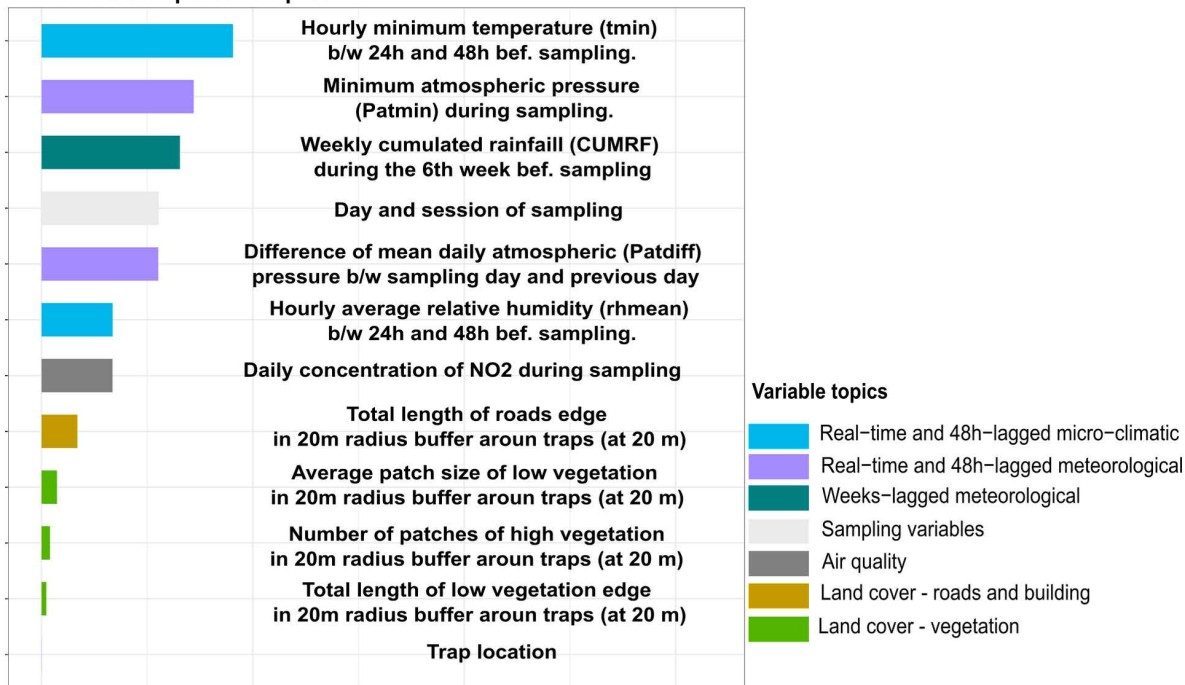

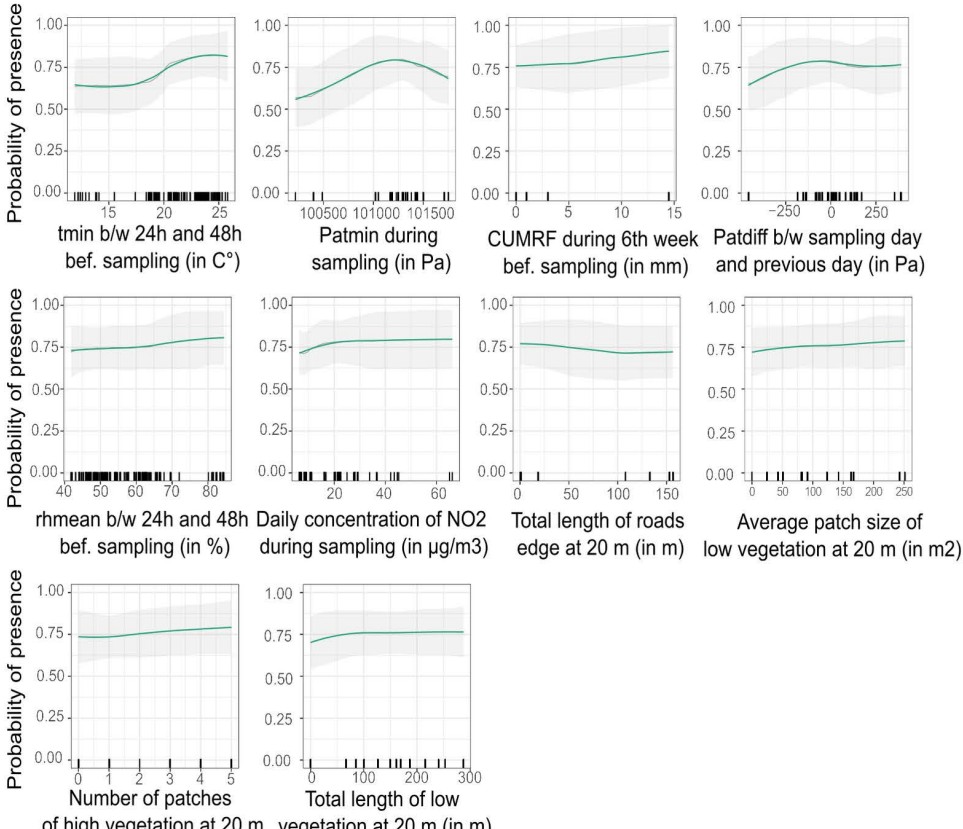

**Fig 5. Interpretation plots for the multivariate random forest model of *Ae. albopictus* presence.** The left plot (Fig 5A) is the variable importance plot (VIP). Rights plots (Fig 5B) are smoothed partial dependence plots (PDPs) with Confidence Intervals for each selected variable in the models. The

y-axis in PDPs represents the probability of catching at least one mosquito. The range of values in the x-axis represents the range of values available in the data for the considered variable. The rugs above the x-axis represent the actual values available in the data for the variable. The black line represents the unsmoothed PDP, while the green line shows the smoothed version. The shaded gray area corresponds to the 95% confidence intervals, calculated using the standard deviation and adjusted for local data density. b/w: between. bef.: before. tmin: minimum hourly temperature. rhmean: hourly average relative humidity. Patmin: minimum daily atmospheric pressure. Patdiff: difference of mean daily atmospheric pressure. CUMRF: Cumulated Rainfall.

markedly for areas between 0 and 120 m2, plateaued up to 237 m2, and subsequently decreased slightly (Fig 6B). The second most important variable was the local maximum hourly temperature during sampling. A strong positive correlation with abundance was observed between 24°C and 38.7°C, followed by a negative correlation beyond 40.6°C. The third key variable was the total length of road edges within a 250 m buffer around traps, with a strong negative correlation between 8230 m and 11 000 m. The fourth contributor was the percentage of low vegetation within a 250 m buffer, which showed a positive effect between 10% and 23%, beyond which it had no observable effect. The fifth and sixth contributors were the cumulated rainfall in the sixth week before sampling and the weekly GDD between the sampling day and two weeks prior, both of which were positively associated with mosquito abundance. The average wind speed between the day of sampling and five weeks before had no effect on abundance between 3.7 m/s and 4.3 m/s, but showed a negative correlation beyond this range. Finally, the last two contributors were the average patch size of roads within a 250 m buffer and the maximum hourly relative humidity during the 24 hours preceding sampling, both of which were slightly negatively correlated with mosquito abundance.

The abundance model demonstrated good overall accuracy, except for high abundance cases (>11 mosquitoes, Mean Absolute Error (MAE)=8.00, S8A Fig). Spatial cross-validation retained overall abundance trends, except in RES-LEM and PRK-AIG, and showed an underestimation of high abundance and an overestimation of low abundance. Temporal cross-validation, using leave-sampling-session-out, did not preserve abundance trends but captured relative differences in *Ae. albopictus* abundance between areas (S8B Fig).

## Discussion

This study examined, for the first time in France, the impact of urban vegetation on the presence and abundance of *Ae. albopictus*, showing a minimal influence on the presence and a positive association with the abundance. The analytical methodology employed in this study provides a nuanced evaluation of how different environmental factors contribute to mosquito presence and abundance. It highlights the existence of non-linear relationships, offering valuable insights for identifying effective vector control measures.

### Influence of urban vegetation and other land cover variables

This study highlights that urban vegetation within a 20 m buffer radius around traps had a minimal weakly positive influence on the probability of the presence of *Ae. albopictus*. Regarding the abundance, urban vegetation demonstrated a positive effect and emerged as one of the four most influential predictors. This positive effect of urban vegetation on the abundance of *Ae. albopictus* has also been observed in other countries, including Italy [26], Brazil [88], Singapore [89], and China [33]. In our study, the first vegetation metric influencing abundance was the average size of low-vegetation patches within a 50 m buffer radius. These findings are consistent with those of Cianci *et al* (2015) [26], who reported a positive relationship between vegetation cover within a 50 m radius buffer around traps and *Ae. albopictus* eggs abundance in Italy. Manica *et al* (2016) observed that, in Roma (Italy), vegetation cover within a 20 m radius around traps positively influenced *Ae. albopictus* female abundance [27]. They hypothesised that the tiger mosquito prefers "small green islands". Our findings refine this hypothesis by demonstrating that *Ae. albopictus* abundance significantly increases with

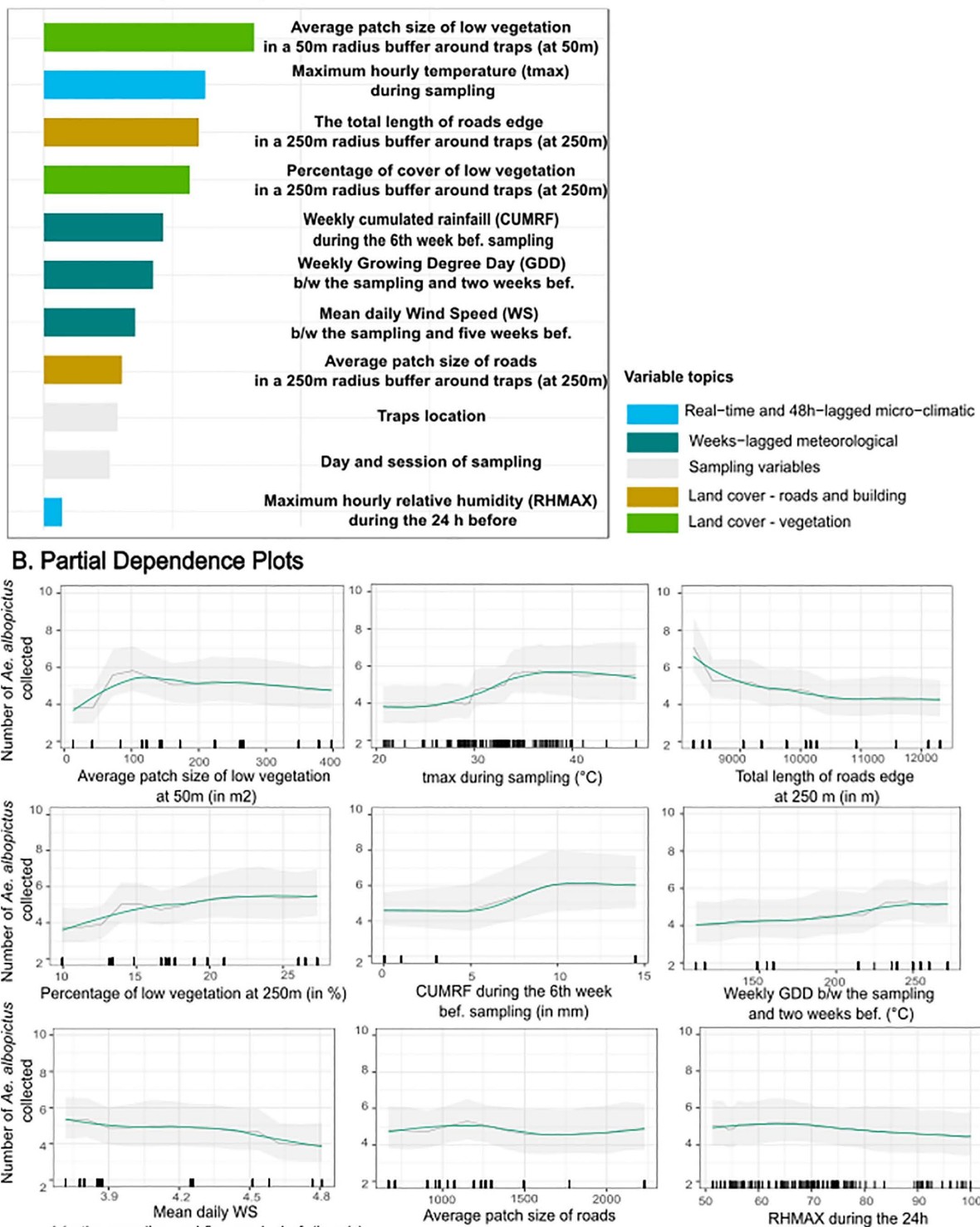

**Fig 6. Interpretation plots for the multivariate random forest model of *Ae. albopictus* abundance.** The left plot (Fig 6A) is the variable importance plot (VIP). Rights plots (Fig 6B) are smoothed partial dependence plots (PDPs) with Confidence Intervals for each selected variable in the models. The y-axis in PDPs represents the number of *Ae. albopictus* females caught/trap/24h. The range of values in the x-axis represents the range of values

available in the data for the considered variable. The rugs above the x-axis represent the actual values available in the data for the variable. The black line represents the unsmoothed PDP, while the green line shows the smoothed version. The shaded gray area corresponds to the 95% confidence intervals, calculated using the standard deviation and adjusted for local data density. b/w: between. bef.: before. tmax: maximum hourly temperature. GDD: Growing Degree Day. WS: Wind Speed.

the average size of low-vegetation patches, particularly within the 10–120 m2 range. Beyond 120 m2, the effect stabilises. This nuance provides additional detail to the assumption proposed by Manica *et al*. This hypothesis, along with the identified size range, requires further research to confirm its generalisability across different urban contexts and mosquito populations.

The percentage of low vegetation cover within a 250 m radius around traps moderately increased abundance. These results are consistent with the study of Sun *et al* (2021), which demonstrated that, in Singapore, *Ae. albopictus* abundance increased with managed vegetation within a 300 m radius buffer around traps [89]. However, Manica *et al* (2016) found a negative relationship between vegetation cover within a 300 m radius buffer and mosquito abundance [27]. These differences can be attributed to variations in city architecture and study design. In our study, the percentage of low vegetation cover ranged from 10% to 23.5%, whereas in Manica *et al*, it ranged from 30% to 55%. With different ranges of vegetation cover, differing study designs, and contrasting urban environments, direct comparisons are challenging.

The other two land cover variables, the third and eighth contributors to abundance, are road metrics within a 250 m radius buffer around traps. The first was the total length of road edges, which was negatively correlated with abundance. These findings aligned with the study by Wang *et al* (2023) in Shanghai (China) which reported a negative correlation between road length and *Ae. albopictus* eggs density [33]. This supports our initial assumption that roads act as barriers to mosquito movement. This hypothesis was also consistent with Regilme *et al* (2021), who observed genetic differentiation in *Ae. aegypti* populations located north and south of a major road in Shanghai [42]. The second road-related variable was the average size of road patches. Roads may represent unfavorable habitats for host-seeking *Ae. albopictus* which could explain their limited abundance in areas dominated by large roads.

## Influence of microclimatic and meteorological variables

The relationship between atmospheric pressure and the presence of *Ae. albopictus* in the field has, to our knowledge, been investigated for the first time in this study. The minimum atmospheric pressure during the 24-hour sampling period and the difference in daily mean atmospheric pressure between the sampling day and the previous day were the second and fourth main variables contributing to presence. Atmospheric pressure can influence insects' flight [51] and locomotor activity [52], as shown in *Drosophila* and beetles. Haufe (1954) demonstrated that atmospheric pressure affects the flight activity of *Ae. aegypti*, especially the change in atmospheric pressure after six hours of acclimatisation [53]. A study by Knop *et al* (2023) in Switzerland showed that the traffic rate of high-flying insects (number of insects per hour) had a positive correlation with air pressure [90]. Our results suggest an effect of the atmospheric pressure metrics on the activity and behaviour of the tiger mosquito. Further studies under controlled laboratory and field conditions are needed to investigate these findings in more detail.

Local hourly temperatures emerged as the primary determinant of *Ae. albopictus* presence and the second most influential predictor of its abundance. Evans *et al* (2019) found an exponential relationship between the minimum temperature during the 7 days preceding sampling and the abundance of females, as well as an unimodal relationship between the maximum temperature and the abundance of females, with an optimal daily maximum temperature around 35°C [32]. Our results suggest that increasing daily maximum temperature increases the activity of *Ae. albopictus* up to 38°C.

The third variable influencing the presence of *Ae. albopictus* was the weekly cumulated rainfall recorded six weeks before sampling, which also ranked as the fifth most important predictor of abundance. While other studies have identified

a positive relationship between *Ae. albopictus* dynamics and rainfall accumulated over several weeks before sampling, none have specifically examined a six-week delay. Roiz *et al* (2015) observed in Montpellier that strong rainfall, particularly that accumulated each week over the four weeks before sampling, positively influenced abundance [31]. Similarly, Manica *et al* (2016) reported a positive relationship between the cumulated rainfall one month before sampling and mosquito abundance [27]. The positive effect of rainfall in our study may be explained by the filling of larval habitats, benefiting previous mosquito generations that influenced the sampled population. This hypothesis requires confirmation through additional temporal replicates to validate the observed trend.

The sixth most influential variable of abundance was the weekly GDD during the two weeks preceding sampling. Based on minimum, maximum, and average temperatures, the GDD is widely used as a key predictor for many insect species [91]. For *Ae. albopictus*, the GDD is calculated using a baseline temperature of 11°C, promoting mosquito development when temperatures exceed this value [50]. This baseline corresponds to the average annual temperature required for the spread of *Ae. albopictus* [92].

One of the last variables influencing abundance is the mean daily wind speed during the five weeks preceding sampling. Le Goff *et al* (2024) observed a similar negative relationship between monthly average daily wind speed and *Ae. albopictus* eggs abundance on the Réunion Island [93]. Adeleke *et al* (2022) reported that in regions where the tiger mosquito is present, wind speeds exceeding 4 m·s$^{-1}$ reduce the likelihood of occurrence [94]. The threshold identified in our study (4.3 m·s$^{-1}$) can be considered comparable to the findings of Adeleke *et al*, although direct comparisons between the studies are challenging due to differences in spatial and temporal scales.

## Limitations, perspectives and conclusion

In order to increase the resilience of Urban Green Infrastructures (UGIs) [95] and to adapt the UGIs management to promote biodiversity [96], since 2017, studies have highlighted the need to consider UGIs as socio-ecological ecosystems. This reflects the complexity of urban greening, which must account for land cover, socio-demographic variables, and urban architecture, as demonstrated in the systematic review by Menconi *et al* (2021) [97]. The challenge of studying the relationship between urban greening and the risk of vector-borne diseases is one of the key conclusions of the scoping review by Mercat *et al* (2025) [24]. The difficulty stems from the need to integrate multiple variables essential to the analysis of this socio-ecosystem, the time required to observe its effects [28], and the challenge of assessing vector-borne disease risk using standardised protocols.

Due to limited financial and human resources, this study was limited to specific sampling areas within vegetated environments representative of the city's UGS and non-vegetated environments. A randomised sampling approach would have better captured the diversity of land cover categories. The predicted abundance of *Ae. albopictus* in the RES-SOUL residential area was therefore certainly underestimated. While the abundance model effectively captures temporal variation, it does not account well for spatial variation. This suggests that key factors such as environmental carrying capacity, defined as the 'equilibrium density of larvae in a system' [98], and the availability of resting sites may have been overlooked [35]. This is also related to the complexity of describing urban vegetation using remote sensing [24]. In our study, we employed a fine-scale characterisation of urban vegetation generated through deep learning and photo interpretation, alongside a surface modelling metric derived from lidar data. However, this approach does not capture the diversity of plant species or genera, which can vary markedly between the Botanical Garden, the Park of Aiguelongue, and residential areas. Finally, as described in the Materials and Methods, no municipal vector control interventions were observed in the sampling areas. Nevertheless, potential actions by residents in houses close to the sampled sites could not be assessed.

The objectives of this study were to explore the relationships between urban vegetation and other environmental factors with the presence and abundance of *Ae. albopictus*, to generate hypotheses using an exploratory approach. The realisation of a predictive model is another objective, which requires a better understanding of the ecological and biological mechanisms [99]. The results of the study are based on only one sampling year. To draw more robust conclusions,

additional years of sampling would be needed to assess inter-annual variation. In addition, this study has focused solely on one aspect of the vector risk, the presence and abundance of the vector, without considering its capacity to transmit pathogens, or human exposure and vulnerability. The vector's capacity depends on the abundance of females, the daily survival rate, and the extrinsic incubation period of pathogens [100]. It would be relevant to investigate the effect of urban vegetation on these elements. Together with these investigations, a socio-ecological approach is essential to gain a comprehensive understanding of vector-borne disease risks in urban green spaces. Developing social surveys to assess citizens' perceptions, alongside employing co-construction methods [101] with policymakers, urban designers, and ecologists, could help formulate effective recommendations to mitigate this risk. One example could be the development of green wind corridors, as implemented in Stuttgart [102] and Lisbon [103], which can reduce urban heat islands and increase wind speed, limiting mosquito abundance. Another possible implication for urban planning would be to promote the establishment of small vegetation patches located at least 50 m away from residential areas. These patches could be separated by permeable soil or arranged as discontinuous corridors composed of diverse vegetation types (e.g., grass, scattered trees). Vegetation management and good gardening practices could further limit mosquito proliferation, for instance by emptying potential breeding sites at least once a week, avoiding plants that naturally serve as larval habitats (e.g., bromeliads [104]), and reducing features such as bamboo nodes [29]. Maintaining less dense shrubs could help reduce the number of available resting sites [35]. Such practices would allow urban green spaces to retain their ecological and recreational functions while reducing mosquito abundance. However, these suggestions are currently limited to the local context of Montpellier and would require additional sampling, including in other locations, to ensure more robust and generalizable recommendations.

Urban greening appears to be a promising solution to address the environmental and health challenges associated with increasing urbanisation. It offers numerous health and environmental benefits, but can also contribute to an increased risk of vector-borne diseases. This study, the first in France to investigate the effect of urban vegetation on *Ae. albopictus*, provides a nuanced assessment of how urban vegetation influences both the presence and abundance of mosquitoes. Our results indicate that urban vegetation moderately increases abundance, but has no effect on presence. Urban greening is a dynamic and complex phenomenon that requires a social-ecological approach in order to be studied. Further research is needed to better elucidate the role of urban vegetation in the ecology of the tiger mosquito and its influence on different aspects of risk.

## Supporting information

**S1 Table. Table with sampling areas characterization.**
(DOCX)

**S2 Table. Summary of the different variables used, with their processing, acquisition methods or source database and temporal/spatial scale.**
(DOCX)

**S3 Table. Description of the landscape metrics for each landscape class for each trap, by buffer.**
(DOCX)

**S4 Table. Mosquito density/trap/24h by species and sex in the nine different sampling areas.** The mosquito density represents the marginal mean of the number of mosquitoes caught per trap per 24 hours, calculated based on the sampling environment, with the standard error specified (SE).
(DOCX)

**S5 Table. Variables selected after bivariate analyses, with justifications for exclusion from multivariate analysis (correlation, VIF, recursive feature elimination).**
(DOCX)

**S1 Fig. BG-PRO trap on the field and pictures of studied areas.** S1A Fig represents a BG-Pro on the field, S1B Fig the Park of Aiguelongue, S1C the Botanical Garden, S1D The Bouisson Bertrand Institute and 1E the Acapulco Hostel. (TIFF)

**S2 Fig. Analysis workflow: From the data acquisition to the multivariate analysis.** (TIF)

**S3 Fig. Time series of weekly cumulated precipitations (1A), mean (TMEAN), minimum (TMIN) and maximum (TMAX) temperature, (1B), growing degree-days (1B), mean relative humidity (1C), mean wind speed (1D) and mean (Patmean), minimum (Patmin) and maximum atmospheric pressure (Patmax), during the studing period.** Vertical dotted red lines indicate monthly mosquito sampling sessions. (TIFF)

**S4 Fig. Number of *Ae. albopictus* per trap per 24h during the study period, by sampling environment.** Boxes represent medians and interquartile ranges, whiskers indicate minimum and maximum values, and points show the abundance for each sampling session and day, for each trap, within the three environments. (TIFF)

**S5 Fig. Cross correlation maps showing the lagged effect of air-quality variables on *Ae. albopictus* presence or abundance.** Lagged air-quality variables include daily concentration of nitrogen monoxide (NO), of nitrogen dioxide (NO2), of particulate matter (PM10, PM2.5) and of ozone (O3). Time lags are expressed in weeks prior to sampling. The adjusted marginal R2 reflects the variance explained by the explanatory variable, adjusted for correlation direction. Red-bordered squares highlight the time lag with the highest marginal R2, with the interval indicated in the top left corner. Black-bordered squares denote marginal R2 values close to the highest (within 10%). Gray squares represent correlations with p-values > 0.2. (TIFF)

**S6 Fig. Bivariate relationships between *Ae. albopictus* presence or abundance and the sampling environment and the socio-demographic variables.** The adjusted marginal R2 reflects the variance explained by the explanatory variable, adjusted for correlation direction. Boxes are colored if the p-value was < 0.2 (No asterisk: p-value ∈ [0.05; 0.2], *: p-value ∈ [0.01, 0.05], **: p-value ∈ [0.001; 0.01]; ***: p-value ∈ [0; 0.001]). Box color depends on the direction of the relationship (blue: negative, red: positive). Color intensity varies according to the marginal R2 value. (TIFF)

**S7 Fig. Evaluation of presence model.** S7A Fig is the Receiver Operating Characteristics (ROC) Curve for the presence model. It illustrates the model's sensitivity (its ability to predict the presence of *Ae. albopictus*) against 1 – Specificity (its ability to detect the absence of *Ae. albopictus*). The Area Under the Curve (AUC) represents the model's ability to discriminate between the presence and absence of the mosquito, with values ranging from 0 to 1. The closer the AUC is to 1, the better the model's predictive performance. S7A Fig shows that the model demonstrated good predictive accuracy, with an AUC of 0.89. S7B Fig compares the observed (in green) versus predicted presence probabilities for each out-of-sample leave-area (yellow) and for each out-of-sample leave-sampling-session (in blue). The y-axis represents the probability of *Ae. albopictus* presence, while the x-axis corresponds to the sampling session. Fig 7B indicates that the model accurately predicted the spatiotemporal trends of presence/absence, although it tended to overestimate presence. (TIFF)

**S8 Fig. Evaluation of abundance model.** S8A Fig is a violin plot showing the distribution of residuals for the abundance model. The black dots indicate the median values. The Mean Absolute Error (MAE) and the number of observations (n) are displayed in a square above the plot. This figure demonstrates that the model tends to overestimate low counts of

*Aedes albopictus* (fewer than three caught in 24 hours per trap), accurately estimates counts between 4 and 20, and underestimates higher counts (more than 20). S8B Fig compares the observed versus predicted numbers of *Ae. albopictus* caught per site per sampling session (in green), versus predicted numbers for each out-of-sample leave-area (yellow) and for each out-of-sample leave-sampling-session (in blue). The y-axis represents the mean number of d *Ae. albopictus* females caught per trap per area per sampling session, while the x-axis represents the sampling session. This figure confirms the observations made in S8A Fig.
(TIFF)

**S1 Text. Detailed description of bivariate and multivariate analysis.**
(DOCX)

## Acknowledgments

We would also like to thank Gilbert Le Goff, Nil Rahola, Christophe Paupy for the help to the design sampling and Mathilde Mercat, Coralie Grail and all the other field and laboratory staff for their strong commitment to the V2MOC project. We would like to thank the managers and staff of the Botanical Garden who facilitated our access, the managers of the green spaces of the city of Montpellier for access to the Parc de l'Aiguelongue, as well as the managers of the Insituts Diderot, Bouisson Bertrand, the Hotel Acapulco and the Saint Charles site of the University of Montpellier III, as well as the individuals who gave us access to their gardens. We thank all the persons who worked on the open access databases used in this work.

## Author contributions

**Conceptualization:** Colombine Bartholomée, Emilie Bouhsira, Florence Fournet, Nicolas Moiroux.

**Data curation:** Colombine Bartholomée, Paul Taconet, Nicolas Moiroux.

**Formal analysis:** Colombine Bartholomée, Paul Taconet, Nicolas Moiroux.

**Funding acquisition:** Florence Fournet.

**Investigation:** Colombine Bartholomée, Mathilde Mercat, Coralie Grail.

**Methodology:** Colombine Bartholomée, Paul Taconet, Nicolas Moiroux.

**Project administration:** Florence Fournet.

**Resources:** Florence Fournet.

**Supervision:** Emilie Bouhsira, Florence Fournet, Nicolas Moiroux.

**Validation:** Colombine Bartholomée, Paul Taconet, Nicolas Moiroux.

**Visualization:** Colombine Bartholomée, Paul Taconet.

**Writing – original draft:** Colombine Bartholomée, Nicolas Moiroux.

**Writing – review & editing:** Paul Taconet, Emilie Bouhsira, Florence Fournet.

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
