## [Decision Letter · Decision Letter 0]

1 Sep 2025

PONE-D-25-17422Investigating the role of urban vegetation alongside other environmental variables in shaping Aedes albopictus presence and abundance in Montpellier, FrancePLOS ONE

Dear Dr. Bartholomée,

Thank you for submitting your manuscript to PLOS ONE. After careful consideration, we feel that it has merit but does not fully meet PLOS ONE’s publication criteria as it currently stands. Therefore, we invite you to submit a revised version of the manuscript that addresses the points raised during the review process.

We look forward to receiving your revised manuscript.

Kind regards,

Rajib Chowdhury, M.Sc.; MPH

Academic Editor

PLOS ONE

Journal Requirements:

2. Please note that PLOS ONE has specific guidelines on code sharing for submissions in which author-generated code underpins the findings in the manuscript. In these cases, we expect all author-generated code to be made available without restrictions upon publication of the work. 

Please review our guidelines at https://journals.plos.org/plosone/s/materials-and-software-sharing#loc-sharing-code and ensure that your code is shared in a way that follows best practice and facilitates reproducibility and reuse.

“This work is part of the V2MOC project,  led by FF and funded by the Occitanie Region as part of the Défi Clé RIVOC (https://www.umontpellier.fr/universite/projets-emblematiques/defis-cles-de-la-region-occitanie). CB received a doctoral scholarship from the Défi Clé RIVOC of the Occitanie Region and the University of Montpellier.”

4. We noted in your submission details that a portion of your manuscript may have been presented or published elsewhere:

“Field data have been published on GBIF: https://www.gbif.org/dataset/8e52f35a-2522-4865-8361-5c249310a7cf”

5. Please note that your Data Availability Statement is currently missing the repository name and direct link to access each database. If your manuscript is accepted for publication, you will be asked to provide these details on a very short timeline. We therefore suggest that you provide this information now, though we will not hold up the peer review process if you are unable.

6. We note that Figure 1 in your submission contain map/satellite images which may be copyrighted. All PLOS content is published under the Creative Commons Attribution License (CC BY 4.0), which means that the manuscript, images, and Supporting Information files will be freely available online, and any third party is permitted to access, download, copy, distribute, and use these materials in any way, even commercially, with proper attribution. For these reasons, we cannot publish previously copyrighted maps or satellite images created using proprietary data, such as Google software (Google Maps, Street View, and Earth). For more information, see our copyright guidelines: http://journals.plos.org/plosone/s/licenses-and-copyright.

1) You may seek permission from the original copyright holder of Figure 1 to publish the content specifically under the CC BY 4.0 license.  

2) If you are unable to obtain permission from the original copyright holder to publish these figures under the CC BY 4.0 license or if the copyright holder’s requirements are incompatible with the CC BY 4.0 license, please either i) remove the figure or ii) supply a replacement figure that complies with the CC BY 4.0 license. Please check copyright information on all replacement figures and update the figure caption with source information. If applicable, please specify in the figure caption text when a figure is similar but not identical to the original image and is therefore for illustrative purposes only.

Reviewers' comments:

Reviewer's Responses to Questions

**Comments to the Author**

1. Is the manuscript technically sound, and do the data support the conclusions?

Reviewer #1: Yes

Reviewer #2: Partly

2. Has the statistical analysis been performed appropriately and rigorously? 

Reviewer #1: Yes

Reviewer #2: Yes

3. Have the authors made all data underlying the findings in their manuscript fully available?

Reviewer #1: Yes

Reviewer #2: Yes

4. Is the manuscript presented in an intelligible fashion and written in standard English?

Reviewer #1: Yes

Reviewer #2: Yes

5. Review Comments to the Author

Reviewer #1: Dear Authors,

this work needs to be a great efforts during the data collection and evaluation of the collected data. manuscript techically sound but not easy to follow during reading. therefore need to be shortened some parts for easy to follow. maybe PCA is another option for the evaluating of parameters contirubion presence and abundance.

Reviewer #2: Overall, this is an interesting and timely study examining how urban vegetation affects the presence and abundance of Aedes albopictus in Montpellier, France. The topic is important, bridging urban planning, vector ecology, and public health. The research objectives are clearly stated, and the study provides useful insights that could inform future management of urban green spaces and vector control strategies.

Technical Soundness and Data Support

The study is generally sound, and the data collected are extensive. The methodology is mostly robust, but some clarifications would help fully support the conclusions:

The exact temporal lags used for meteorological and environmental variables should be specified more clearly.

It is unclear whether datasets with different spatial resolutions (e.g., land cover, population, infrastructure) were harmonised or simply intersected with trap locations and buffers.

No information is provided on potential vector control interventions, such as municipal insecticide spraying or larvicide treatments, which could affect mosquito presence and abundance. Clarifying whether these were monitored or considered in the analysis would strengthen the study.

Statistical Analysis

The statistical approach appears rigorous and appropriate. The two-stage modelling strategy—using GLMMs for bivariate screening and random forest models for multivariate analysis—is well justified and accounts for potential non-linear relationships and interactions. Including both spatial and temporal cross-validation adds robustness.

However, the supplementary materials could include more details to enhance reproducibility and transparency, such as: Specifications of the GLMM and random forest models (e.g., distributions, link functions, hyperparameters), Cross-validation results and model diagnostics, Residual spatial autocorrelation analyses, Sensitivity analyses regarding buffer sizes and temporal lags.

Presentation and Writing

The manuscript is generally clear and written in standard English. The structure is logical, and the main concepts are understandable. Minor improvements could make it even smoother:

-Improve transitions in the Introduction between urbanisation and re-greening.

-Clarify certain phrases in Materials & Methods (e.g., “were vegetated” → “were vegetated areas”; precise description of 24- vs 48-hour temporal lags; details of microclimate and rainfall measurements).

-Ensure consistent definitions for land cover metrics and statistical indices.

Additional Comments and Recommendations

The Discussion is strong and compares findings with previous studies well, but it could more explicitly inform practical urban planning, such as optimal green area dimensions, vegetation management strategies, and balancing ecological benefits with vector-related risks.

Recording vegetation type (plant species) in future studies could help identify which species influence mosquito presence and abundance, providing actionable guidance for urban planners.

It would be valuable in future studies to examine whether arbovirus cases occurred in areas corresponding to environmental variables positively or negatively associated with mosquito abundance to reinforce the public health relevance.

Since the study is based on a single year of sampling, additional years of data would allow more robust conclusions.

Conclusion

This study makes a valuable contribution to understanding urban vector ecology and the role of vegetation in shaping mosquito populations. With clarifications and additional details on statistical methods and environmental factors, the manuscript could provide strong, actionable insights for urban planning and vector management.

A more detailed file with comments are attached to this review.

6. PLOS authors have the option to publish the peer review history of their article (what does this mean? ). If published, this will include your full peer review and any attached files.

**Do you want your identity to be public for this peer review?** For information about this choice, including consent withdrawal, please see our Privacy Policy .

Reviewer #1: No

Reviewer #2: No

---

## [Author Response · Author response to Decision Letter 1]

2 Oct 2025

A detailed, point-by-point response with clearer formatting and full details is provided in the file entitled “Response to Reviewers.” We confirm that all requested revisions have been implemented in the revised manuscript.

During the revision process, we also identified an error, which has been corrected. This correction is described in the “Additional Changes” section below and detailed in the accompanying response letter.

For convenience, and as requested, we have also copied below the reviewer and editor comments together with our responses.

## General answer

In response to your suggestions, we made several important revisions. Specifically, we clarified and expanded the Materials and Methods section, incorporated additional discussion points on urban greening, and updated the Supporting Information files to improve transparency and reproducibility. Figures 1, 2, 3, and 6 were revised in line with PLOS ONE guidelines and copyright policies, with updated legends providing more comprehensive detail.

During a review of spatial residual autocorrelations, we also identified an error in the multivariate abundance models. This issue has now been corrected in the Results and Discussion sections. Although the correction modified the ranking in the variable importance plots, it did not alter the overall interpretation or conclusions of the study.

In compliance with the editorial requirements, we also made the following clarifications:

• Financial disclosure: The funders had no role in study design, data collection and analysis, decision to publish, or preparation of the manuscript.

• GBIF dataset: The dataset shared on GBIF (https://www.gbif.org/dataset/8e52f35a-2522-4865-8361-5c249310a7cf or https://doi.org/10.15468/4qafbu) consists solely of raw, open-access occurrence records of Aedes albopictus collected during our field surveys. These data were not peer-reviewed and have not been formally published in any journal or conference proceedings. The GBIF submission serves only as a public repository to support open data practices, and does not constitute prior publication of the analyses, interpretations, or conclusions presented in this manuscript.

• Data and code availability: All data are available through the above GBIF repository (https://www.gbif.org/dataset/8e52f35a-2522-4865-8361-5c249310a7cf or https://doi.org/10.15468/4qafbu) , and the statistical analysis code is provided at (https://archive.softwareheritage.org/swh:1:dir:a1c83c4be076b5645d1834265219d4f2e6e5b544;origin=https://github.com/ptaconet/modeling_vector_mtp;visit=swh:1:snp:741ed0f1af295a2a2ef477b310cda3f0faa3f1a0;anchor=swh:1:rev:11de6e23243904d7fa7a21d9d6ade29486d06349).

• Figure 1: This figure has been replaced with an original image created by the authors, fully compliant with the CC BY 4.0 license.

This response package is organized as follows: (1) a summary of additional corrections made to the manuscript, (2) clarifications in response to journal-specific requirements, and (3) point-by-point replies to the comments from Reviewer 1 and Reviewer 2.

## Additional changes

When assessing spatial residual autocorrelation in the Random Forest abundance models (Reviewer 2, Comments 4 & 18), we discovered that, during recursive feature elimination, some variables essential for accounting for autocorrelation had been excluded. We therefore reintroduced these variables into the final model (namely, cumulative rainfall during the six weeks preceding sampling and maximum hourly relative humidity within the 24 hours prior to sampling). This adjustment altered the ranking in the variable importance plot for abundance but did not change the overall interpretation of the results. Accordingly, we revised:

- the Abstract:

o Lines 40-42: “While urban vegetation had a limited effect on Ae. albopictus presence, the average patch size, and the percentage of area covered by low vegetation were among the most important predictors of abundance.”

o Lines 44-46: “The most important predictors of abundance were the average patch size of low vegetation, the maximum hourly temperatures during sampling, and the length of roads.”

- the Results section, specifically the “Abundance Model” paragraph within “Multivariate Analysis”

o Lines 560-580:”The most contributing variables to Ae. albopictus abundance comprised a combination of microclimatic, weeks-lagged meteorological, vegetation and other land cover variables. The first contributing variable was the average size of low-vegetation patches within a 50 m radius buffer around traps (Fig 6A). Abundance increased markedly for areas between 0 and 120 m², plateaued up to 237 m², and subsequently decreased slightly (Fig 6B). The second most important variable was the local maximum hourly temperature during sampling. A strong positive correlation with abundance was observed between 24°C and 38.7°C, followed by a negative correlation beyond 40.6°C. The third key variable was the total length of road edges within a 250 m buffer around traps, with a strong negative correlation between 8230 m and 11 000 m. The fourth contributor was the percentage of low vegetation within a 250 m buffer, which showed a positive effect between 10% and 23%, beyond which it had no observable effect. The fifth and sixth contributors were the cumulated rainfall in the sixth week before sampling and the weekly GDD between the sampling day and two weeks prior, both of which were positively associated with mosquito abundance. The average wind speed between the day of sampling and five weeks before had no effect on abundance between 3.7 m/s and 4.3 m/s, but showed a negative correlation beyond this range. Finally, the last two contributors were the average patch size of roads within a 250 m buffer and the maximum hourly relative humidity during the 24 hours preceding sampling, both of which were slightly negatively correlated with mosquito abundance. The abundance model demonstrated good overall accuracy, except for high abundance cases (>11 mosquitoes, Mean Absolute Error (MAE)=8.00, S8A Fig).’

o Figure 6 has been updated to reflect the revised results and has been uploaded as part of the revised submission.

- the Discussion section

o Lines 587 and 589: “This study examined, for the first time in France, the impact of urban vegetation on the presence and abundance of Ae. albopictus, showing a minimal influence on the presence and a positive association with the abundance”

o Lines “597-598”: “Regarding the abundance, urban vegetation demonstrated a positive effect and emerged as one of the four most influential predictors”

o Lines 600-601: “In our study, the first vegetation metric influencing abundance was the average size of low-vegetation patches within a 50 m buffer radius. “

o Lines 623-624: “The other two land cover variables, the third and eighth contributors to abundance, are road metrics within a 250 m radius buffer around traps. “

o Lines 648 – 649: “Local hourly temperatures emerged as the primary determinant of Ae. albopictus presence and second most influential predictor of its abundance.”

o Lines 655-657; “The third variable influencing the presence of Ae. albopictus was the weekly cumulated rainfall recorded six weeks before sampling, which also ranked as the fifth most important predictor of abundance.”

o Lines 667-668: “The sixth most influential variable of abundance was the weekly GDD during the two weeks preceding sampling.”

o Lines 671 and 674: [50]. “This baseline corresponds to the average annual temperature required for the spread of Ae. albopictus [92]. One of the last variables influencing abundance is the mean daily wind speed during the five weeks preceding sampling.”

- Supplementary Figure S8 has been updated to reflect the revised results and has been uploaded as part of the revised submission.

- The files 6 and 7 related to the code for the multivariate model have been updated, as reflected in the publicly accessible code repository (https://archive.softwareheritage.org/swh:1:dir:a1c83c4be076b5645d1834265219d4f2e6e5b544;origin=https://github.com/ptaconet/modeling_vector_mtp;visit=swh:1:snp:741ed0f1af295a2a2ef477b310cda3f0faa3f1a0;anchor=swh:1:rev:11de6e23243904d7fa7a21d9d6ade29486d06349).

## Journal Requirements

- The manuscript has been carefully reviewed to ensure full compliance with all PLOS ONE style requirements, including file naming conventions, and any necessary adjustments have been made.

- The code has been made available as indicated in the Methods section (last sentence: “All scripts are available on Software Heritage [87].”). Our code is shared in accordance with best practice, facilitating reproducibility and reuse.

- This statement regarding the role of the funders has been added to the cover letter: 'The funders had no role in study design, data collection and analysis, decision to publish, or preparation of the manuscript.'

- The dataset we shared on GBIF represents raw, open-access occurrence records of Aedes albopictus collected during our field surveys. These records were not peer-reviewed and have not been formally published in any journal or conference proceedings. The GBIF submission serves solely as a public data repository to comply with open data practices and does not constitute prior publication of the analyses, interpretations, or conclusions presented in the current manuscript. Therefore, the content of the present manuscript, including statistical analyses, integration with environmental and landscape variables, and ecological interpretation, is entirely original and has not been published elsewhere.

This statement has been added to the the reponse letter.

- We apologize for this oversight. As noted in the manuscript (lines 179, ref. 46), both the entomological data and the primary environmental and climatic datasets are publicly available via GBIF and can be accessed directly at the following link: https://doi.org/10.15468/4qafbu.

- Thank you for your helpful suggestion. In response, we have removed the Google Maps image and replaced it with our own photographs of the study areas. To ensure proper attribution, we have also included a reference to OpenStreetMap in the figure caption: “Map data © OpenStreetMap contributors, licensed under ODbL.” (Lines 151-152).

## Reviewer #1:

Dear Authors, this work needs to be a great efforts during the data collection and evaluation of the collected data. manuscript techically sound but not easy to follow during reading. therefore need to be shortened some parts for easy to follow. Maybe PCA is another option for evaluating the parameters contribution presence and abundance.

We thank Reviewer 1 for the comment regarding manuscript length. We have condensed the text where possible, but certain sections—particularly on the creation of the land cover map and the extraction of socio-economic data (Comment 2)—needed to be expanded to ensure reproducibility, as asked by Reviewer 2. These additions provide necessary detail for transparency and clarity, while the rest of the manuscript remains concise. We thank the reviewer for this interesting suggestion. Principal Component Analysis (PCA) is indeed a valuable method for dimensionality reduction and for summarizing correlated predictors. However, in our study, the aim was not to reduce predictors into composite axes but rather to evaluate the direct contribution of specific environmental and landscape variables (e.g., vegetation cover, building density, meteorological conditions) to the presence and abundance of Ae. albopictus. Using PCA would have made the interpretation of individual variable effects more difficult, since biological meaning is less straightforward once predictors are combined into principal components.

To address multicollinearity among predictors, we instead applied variance inflation factor (VIF). This allowed us to retain explanatory variables while ensuring model robustness, and at the same time to quantify the effect sizes of specific predictors that are directly relevant to mosquito ecology and urban planning.

## Reviewer #2:

Overall, this is an interesting and timely study examining how urban vegetation affects the presence and abundance of Aedes albopictus in Montpellier, France. The topic is important, bridging urban planning, vector ecology, and public health. The research objectives are clearly stated, and the study provides useful insights that could inform future management of urban green spaces and vector control strategies.

Technical Soundness and Data Support

The study is generally sound, and the data collected are extensive. The methodology is mostly robust, but some clarifications would help fully support the conclusions:

Comment 1: The exact temporal lags used for meteorological and environmental variables should be specified more clearly.

Please see our answer to comments 23-24 & 26.

Comment 2: It is unclear whether datasets with different spatial resolutions (e.g., land cover, population, infrastructure) were harmonised or simply intersected with trap locations and buffers.

We thank the reviewer for this important comment. We confirm that datasets with different spatial resolutions were harmonised prior to analysis. Specifically, we combined three datasets to produce a consistent land-cover map (0.5 m resolution) used in subsequent analyses (see Supplementary Table 2):

- Vegetation dataset: fine-scale vegetation map of Montpellier (Montpellier Méditerranée Métropole), generated using a hybrid approach combining deep learning image processing and photo-interpretation of Pléiades satellite imagery with a LiDAR-derived elevation model. This vector dataset included polygons > 5 m².

- Building dataset: French national building database (Base de données nationale des bâtiments, BDNB), created by cross-referencing ~20 public datasets. This vector dataset has < 1 m precision.

- LULC dataset: Copernicus Urban Atlas (European Union), vector dataset with polygons > 2,500 m².

Land-cover harmonisation workflow:

- All three datasets were rasterised at 0.5 m spatial resolution.

- Empty cells (NA values) in the vegetation raster were filled using values from the building dataset, resulting in a “vegetation + building” layer.

- Remaining NA cells were filled with values from the LULC dataset.

- Land-cover classes were merged into five categories:

o low vegetation (< 3 m height), high vegetation (> 3 m height), from the vegetation dataset,

o buildings, from the buiding dataset,

o roads, and others, from the LULC dataset.

The R code used for this process is openly available (file 1_prepare_data.R in reference 87 of the manuscript).

Spatial analyses were then performed to extract, for each buffer around traps (20–250 m radii), the percentage of area, total edge length, number of patches of each land-cover class, and the Shannon index of landscape diversity.

The manuscript (Methods section, § Fine-scale vegetation and land cover data) was modified to make it clearer:

“A harmonised land-cover map of Montpellier was generated at 0.5 m resolution by combining three complementary datasets (see Supplementary Table S2): (i) the fine-scale vegetation dataset of Montpellier (Montpellier Méditerranée Métropole, 2019), which distinguishes vegetation patches using deep learning and photo-interpretation of Pléiades satellite imagery and a LiDAR-derived elevation model; (ii) the French national building database (Base de données nationale des bâtiments, BDNB); and (iii) the Copernicus Urban Atlas. Following rasterisation and integration of these sources, five land-cover categories were retained (Fig 1B): buildings, roads, low vegetation (< 3 m height), high vegetation (> 3 m height), and “others” (including railways, pathways, industrial areas, and sports facilities).[...]”

Regarding demographic and social data, we used two complementary databases:

- Montpellier fine-scale population dataset: population counts from each sub-municipal IRIS unit (~2,000 inhabitants) were redistributed proportionally to the living area of individual land plots. The resulting dataset provides georeferenced points (X, Y) located at the centre of significant buildings, each associated with a population value (rounded to the nearest whole number).

- Filosofi database (INSEE, 2

---

## [Decision Letter · Decision Letter 1]

16 Oct 2025

Investigating the role of urban vegetation alongside other environmental variables in shaping Aedes albopictus presence and abundance in Montpellier, France

PONE-D-25-17422R1

Dear Dr. Bartholomée,

We’re pleased to inform you that your manuscript has been judged scientifically suitable for publication and will be formally accepted for publication once it meets all outstanding technical requirements.

Kind regards,

Rajib Chowdhury, M.Sc.; MPH

Academic Editor

PLOS ONE

Additional Editor Comments (optional):

Reviewers' comments:

Reviewer's Responses to Questions

**Comments to the Author**

1. If the authors have adequately addressed your comments raised in a previous round of review and you feel that this manuscript is now acceptable for publication, you may indicate that here to bypass the “Comments to the Author” section, enter your conflict of interest statement in the “Confidential to Editor” section, and submit your "Accept" recommendation.

Reviewer #1: All comments have been addressed

Reviewer #2: All comments have been addressed

2. Is the manuscript technically sound, and do the data support the conclusions?

Reviewer #1: Yes

Reviewer #2: Yes

3. Has the statistical analysis been performed appropriately and rigorously? 

Reviewer #1: Yes

Reviewer #2: Yes

4. Have the authors made all data underlying the findings in their manuscript fully available?

Reviewer #1: Yes

Reviewer #2: Yes

5. Is the manuscript presented in an intelligible fashion and written in standard English?

Reviewer #1: Yes

Reviewer #2: Yes

6. Review Comments to the Author

Reviewer #1: Dear Authors,

thanks for kindly responce of all comments from my side and other rewiever. manuscript further advanced after revision

Reviewer #2: (No Response)

7. PLOS authors have the option to publish the peer review history of their article (what does this mean? ). If published, this will include your full peer review and any attached files.

**Do you want your identity to be public for this peer review?** For information about this choice, including consent withdrawal, please see our Privacy Policy .

Reviewer #1: No

Reviewer #2: No

---

## [Editor Report · Acceptance letter]

PONE-D-25-17422R1

PLOS ONE

Dear Dr. Bartholomée,

I'm pleased to inform you that your manuscript has been deemed suitable for publication in PLOS ONE. Congratulations! Your manuscript is now being handed over to our production team.

Kind regards,

on behalf of

Dr. Rajib Chowdhury

Academic Editor

PLOS ONE